

# Radiation statistics of a degenerate parametric oscillator at threshold

Fabian Hassler, Steven Kim and Lisa Arndt

JARA Institute for Quantum Information, RWTH Aachen University, 52056 Aachen, Germany

## Abstract

As a function of the driving strength, a degenerate parametric oscillator exhibits an instability at which spontaneous oscillations occur. Close to threshold, both the nonlinearity as well as fluctuations are vital to the accurate description of the dynamics. We study the statistics of the radiation that is emitted by the degenerate parametric oscillator at threshold. For a weak nonlinearity, we can employ a quasiclassical description. We identify a universal Liouvillian that captures the relevant long-time dynamics for large photon-numbers. We find that the cumulants obey a universal power-law scaling as a function of the nonlinearity. The Fano factor shows a maximum close, but not coinciding, with the threshold. Moreover, we predict a certain ratio of the first three cumulants to be independent of the microscopic details of the system and connect the results to experimental platforms.



# 1 Introduction

An oscillator that is parametrically driven at twice its natural frequency $\Omega$ exhibits an instability as a function of the driving strength [1–3]. Above the threshold, spontaneous oscillations set in. The amplitude of oscillation increases with a fractional power $1/m$ of the distance from the instability threshold. Generically, the amplitude grows as a square-root (pitchfork bifurcation with $m = 2$). However, for a symmetrical confinement potential in the rotating frame as it arises, *e.g.*, in the case of a Duffing oscillator, it is known that $m = 4$ [4]. What is hidden in the analysis of the stationary state discussed so far, is the critical slowing down close to threshold, *i.e.*, the timescale $\tau_*$ over which the stationary state is reached becomes much larger than the damping time of the system. In fact, the linearized theory predicts a divergence of $\tau_*$ which gets cured by the (weak) nonlinearity of strength $\alpha$. This behavior is a prime example of a bifurcation that exhibits universal behavior and generically appears in driven-dissipative systems [5–8]. Different types of bifurcations lead to a variety of critical exponents and correlation behaviors that have been the focus of many studies in recent years [9–21].

The long timescale becomes particularly important when fluctuations (both quantum or thermal) are included in the description of the parametrically driven system. While these fluctuations have a rather small effect on the behavior well-above threshold where the classical coherent oscillation sets in, it is known that they produce excitations that lead to radiation emitted by the system even below threshold [22–28]. In fact, it has recently been pointed out that the statistics of the resulting radiation collected over a measurement time $\tau \gg \tau_*$ is universal [29]. However, all these theories rely on linearizing the problem and, due to $\tau_* \to \infty$, predict diverging cumulants of the photon counting at threshold.[1] It is well-known that this divergence gets cured by the inclusion of the (weak) nonlinearity. However, obtaining insights into the corresponding statistics at threshold—or more generally in the critical regime around the threshold defined below—is complicated by the fact that both fluctuations and the nonlinearity have to be taken into account. First results were reported in Refs. [23, 30] where the stationary state of the system at threshold was obtained. However, the dynamics and the resulting photon statistics of the system for long measuring times has not been discussed up to now.

In this paper, we derive a universal Liouvillian that accounts for the slow time-evolution (on the scale $\tau_*$) of the system, which is valid in the critical regime for a weak nonlinearity. At threshold, observables show a universal power-law behavior as a function of the nonlinearity $\alpha$. In particular, we show that $\tau_* \propto \alpha^{-2/(m+2)}$ and $N_0 \propto \alpha^{-4/(m+2)}$, where $N_0$ is the number of correlated photons at threshold. The central insight of our work is the fact that, for weak nonlinearities, we can employ a quasiclassical approximation [31–33] as both the relevant timescale as well as the photon number diverge. We test the results by comparing them to numerics solving the Lindblad master equation in the rotating-frame. Our results are valid for a broad range of experimental realizations of parametrically driven oscillators with weak nonlinearities. Some examples are the optical parametric oscillator [23], the Dicke transition [34] of cold-atomic gases in a cavity [8], the optomechanical parametric instability ('mechanical lasing') [35], or the Josephson parametric down-conversion in superconducting circuits [22, 24].

The article is organized as follows. We derive the effective Liouvillian which allows us to identify the characteristic time $\tau_*$ and amplitude at threshold. We provide results for the photon-current and the second-order coherence at $\tau = 0$ which give insights into the stationary state. We proceed by analyzing the dynamical properties. In particular, we calculate the first

---

[1]Refs. [22,28] have finite cumulants without the need of nonlinearities, however they treat a finite measurement time $\tau \lesssim \tau_*$ which cures the divergence.

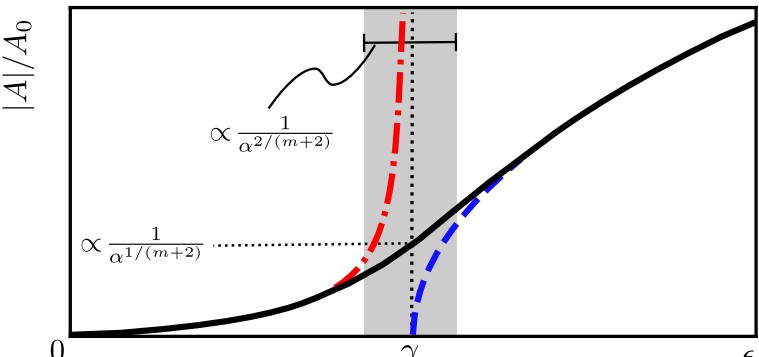

Figure 1: Sketch of the amplitude $A$ of an oscillator undergoing bifurcation at $\epsilon = \gamma$ where the driving exactly cancels the damping. The dashed line indicates the classical solution. Including fluctuations, there is a response even below threshold. In particular, the linearized theory (dash-dotted line) predicts a divergence at threshold. This divergence is cured by the nonlinearity, parameterized by the dimensionless parameter $\alpha$. The present manuscript analyzes the critical regime (gray region) where both fluctuations and the nonlinearity are relevant.

four cumulants of the photon statistics and the second-order coherence at threshold. We finish by comparing the Fano factor of our effective model to numerical results of realistic systems.

## 2 Model

The dynamics of the parametric oscillator is described by the position $X = \mathrm{Re}(A e^{-i\Omega t})$ of the oscillator. For a degenerate oscillator, the slow amplitude $A(t) \in \mathbb{C}$ obeys the $\mathbb{Z}_2$-symmetry $\pm A$. Below threshold, the steady state is characterized by the value $A_s = 0$ while above threshold of bifurcation the amplitude develops a finite classical value $A = \pm A_s$ that corresponds to the spontaneous emergence of a symmetry-breaking oscillation. As we show below, the physics close to threshold is dominated by a long timescale $\tau_*$ (critical slowing-down) and a large amplitude $A_s$, corresponding to many quanta of energy. As a result, it is possible to describe the dynamics of $A$ using a quasiclassical description with

$$\dot{A} = -\frac{\gamma}{2} A + F(A, A^*) + \eta, \tag{1}$$

here, $\gamma$ is the damping rate (due to the coupling to a detector), the force $F$ incorporates the driving, and $\eta$ is a classical, complex noise with zero mean and variance of the Callen-Welton form [31] $\langle \eta(t)^* \eta(t') \rangle = A_0^2 \gamma (\bar{n} + \frac{1}{2}) \delta(t - t')$, with the Bose-Einstein occupation $\bar{n} = (e^{\hbar\Omega/k_B T} - 1)^{-1}$, and $A_0 > 0$ the strength of the zero-point fluctuations.[2] Note that without driving, the noise makes sure that the system on average approaches the stationary value $\langle |A|^2 \rangle = A_0^2(\bar{n} + \frac{1}{2})$ as required by statistical mechanics.

## 3 Universal Liouvillian

For degenerate parametric driving, only one of the two quadratures of the oscillator undergoes a pitchfork bifurcation and, thus, demonstrates a large amplitude. To fix ideas, we assume that

---

[2]For an oscillator of mass $m$, we have $A_0 = \sqrt{2\hbar/m\Omega}$.

the bifurcation is on the real part of $A$ and write $A = A_0(x + iy)/\sqrt{2} \approx A_0 x/\sqrt{2}$, $x, y \in \mathbb{R}$. The real part of (1) follows the Langevin equation

$$\dot{x} = -\frac{\gamma}{2}x + f(x) + \sqrt{\gamma(\bar{n} + \tfrac{1}{2})}\,\xi\,, \tag{2}$$

where $\xi$ is a white noise source with $\langle \xi(t)\xi(t')\rangle = \delta(t - t')$. Due to the symmetry, the force $f(x) = \sqrt{2}\,\mathrm{Re}(F)/A_0$ obeys $f(-x) = -f(x)$. As the nonlinearity is assumed to be weak, we can Taylor expand the force $f(x) \approx \frac{1}{2}(\epsilon x - \gamma \alpha x^{m+1})$;[3] generically, we have $m = 2$, as this is the first term consistent with the $\mathbb{Z}_2$-symmetry. However, we will keep $m$ undetermined as for the important case of a driven Duffing-oscillator, we have $m = 4$ due to the additional rotational symmetry of the confinement potential in the rotating frame, see below. The parameter $\epsilon > 0$ describes the driving strength and $\alpha > 0$ is a dimensionless measure of the nonlinearity, $i.e.$, the nonlinearity at the photon scale. For the following, we need a weak nonlinearity with $\alpha \ll 1$ to ensure that the amplitude $x$ is indeed large at threshold. This allows the quasiclassical description of the driven-dissipative system [31]. Without noise, the system undergoes a classical bifurcation at $\epsilon = \gamma$ with the solution $A_s = 0$ valid for $\epsilon < \gamma$ and $A_s \approx A_0 x_s/\sqrt{2} = A_0[(\epsilon - \gamma)/\alpha\gamma]^{1/m}/\sqrt{2}$ for $\epsilon > \gamma$. In particular, the amplitude above threshold generically increases as $A_s \propto (\epsilon - \gamma)^{1/2}$ (with $m = 2$ [2]) while $A_s \propto (\epsilon - \gamma)^{1/4}$ for the case of a Duffing-oscillator (with $m = 4$ [4]).

As shown in Appendix A, the Langevin equation (2) is equivalent to the Fokker-Planck equation $\dot{P} = \mathcal{L}_0 P$ with the *universal Liouvillian* [4]

$$\mathcal{L}_0 = -\frac{1}{\tau_*}\left(\frac{p^2}{2} + i\beta pq - ipq^{m+1}\right), \tag{3}$$

that is valid close to threshold, where $p, q$ are canonically-conjugate variables obeying $[q, p] = i$ [37, 38]. In this form, the microscopic parameters of the system only enter via the characteristic amplitude $x_* = [(2\bar{n} + 1)/\alpha]^{1/(m+2)}$, with $x = x_* q$ [23], the characteristic timescale $\tau_* = x_*^2/(\bar{n} + \tfrac{1}{2})\gamma \gg \gamma^{-1}$, and the dimensionless distance from the threshold $\beta = \frac{1}{2}(\epsilon - \gamma)\tau_*$. Note that $\mathcal{L}_0$ is an effective model that only describes the slow-dynamics of the system accurately. Because of this, only the eigenvalues with small (negative) real parts are relevant. The description in terms of the universal Liouvillian is only valid for a weak nonlinearity $\alpha \ll 1$ and small temperatures. The temperature is bound by the need for a separation of timescales, $i.e.$, $\gamma\tau_* \gg 1$, which implies $k_B T/\hbar\Omega \lesssim \bar{n} + \frac{1}{2} \ll \alpha^{-1/[(m+3)(m+1)]}$.

The characteristic scales $\tau_*, x_*$ of the system at threshold (with $\epsilon = \gamma$) can be understood in the following way: at threshold, we have the typical force $f_* = f(x_*) \simeq \gamma\alpha x_*^{m+1}$. Fluctuations due to the last term in (2) fix the typical amplitude $x_*$ such that diffusion due to the random force equals the drift due to $f_*$. For this we need, $x_* f_* \simeq \gamma(\bar{n} + \tfrac{1}{2})$ which yields (up to numerical factors) the characteristic amplitude $x_*$ given above. The characteristic time $\tau_*$ follows from (2) as the force $f_*$ sets the velocity $x_*/\tau_* \simeq f_* \simeq \gamma(\bar{n} + \tfrac{1}{2})/x_*$.

## 4 Results

Let us first concentrate on the stationary state of the system without accounting for dynamics. The Liouvillian leads to the stationary distribution $P_s(q) \propto \exp[-2q^{m+2}/(m+2) + \beta q^2]$ that is an eigenstate of $\mathcal{L}_0$ with eigenvalue 0. As a result, the relevant $q$ at threshold are of order 1. This allows to estimate the error due to the next term $\gamma\tilde{\alpha}x^{m+3}$ in the Taylor expansion of

---

[3]We introduce the factor $\gamma$ in the nonlinear term for convenience in order to render $\alpha$ dimensionless.

[4]Note that the universal Liouvillian enjoys a PT-symmetry which guarantees that the spectrum is either real or that the eigenvalues come in complex conjugate pairs, see [36].

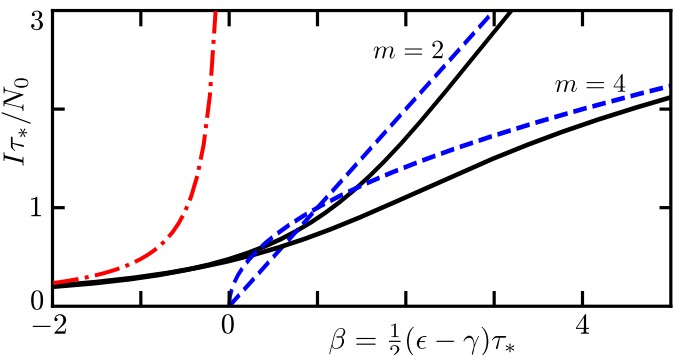

Figure 2: Dimensionless photon current, which is proportional to $A_s^2$, as a function of the driving strength $\beta$, providing a zoom into the critical region of Fig. 1. The solid line is the result (4). It interpolates between the result $1/2|\beta|$ (dash-dotted) for $\beta \to -\infty$ of Refs. [24, 29] and the classical expression $\beta^{2/m}$ (dashed line) for $\beta \to \infty$.

$f(x)$. The relative error scales like $\tilde{\alpha} x_*^2 / \alpha \propto \tilde{\alpha}/\alpha^{(m+4)/(m+2)}$. Typically, systems that show a parametric instability are weakly-coupled with $\tilde{\alpha} \ll \alpha^{(m+4)/(m+2)}$ such that higher-order terms are less relevant and can be neglected. Note that the stationary state at threshold has been discussed in Ref. [23] in the context of optical radiation in a pumped, nonlinear crystal.[5]

We envision that the damping is (partially) due to the coupling of the system to a detector. To characterize the detector, we introduce the photon current $I = f\gamma|A/A_0|^2 \approx \frac{1}{2} f\gamma x^2$ with $0 \le f \le 1$ the counting efficiency. Averaging over the stationary distribution, we obtain the average photon current $\langle I \rangle = \frac{f\gamma x_*^2}{2}\langle q^2 \rangle_s$ which evaluates to (see Fig. 2)

$$\langle I \rangle = \frac{f\gamma x_*^2}{2}\begin{cases} D_{-3/2}(-\beta)/2D_{-1/2}(-\beta), & m=2, \\ 2^{1/3}M'(\beta/2^{2/3})/M(\beta/2^{2/3}), & m=4, \end{cases} \tag{4}$$

here, $D_p(z)$ is the parabolic cylinder function and $M(z) = \sqrt{\text{Ai}(z)^2 + \text{Bi}(z)^2}$ the modulus of the Airy functions [39]. This result describes the behavior in the critical regime connecting the known behavior above and below threshold, see Fig. 1. The second order coherence $g^{(2)}(\tau)$ at vanishing time delay $\tau$ measures the fluctuations in the stationary state. It is given by $g^{(2)}(0) = \langle q^4 \rangle_s / \langle q^2 \rangle_s^2$ and evaluates at threshold to $32\Gamma(5/4)^4/\pi^2 \approx 2.19$ for $m=2$ and 2 for $m=4$.

## 4.1 Counting statistics

More generally, during the detection time $\tau$, the detector measures the statistics of the photon number $N = \int_0^\tau dt\, I$. We can access the counting statistics by adding the source term $\frac{1}{2}sf\gamma x^2$ to the Liouvillian [28, 29]. The eigenvalue $\lambda(\beta, sN_0)/\tau_*$ of

$$\mathcal{L} = \mathcal{L}_0 + \frac{sN_0}{\tau_*}q^2, \qquad N_0 = \frac{f\gamma\tau_* x_*^2}{2}, \tag{5}$$

---

[5]The optical system realizes the generic case $m=2$ with the nonlinearity given by $\alpha \simeq 1/n_p$ where $n_p$ denotes the number of pump photons at threshold without accounting for depletion.

Table 1: Numerical coefficients of the photon counting at threshold in (8).

| $m$ | $c_1$ | $c_2$ | $c_3$ | $c_4$ |
|---|---|---|---|---|
| 2 | $2\pi/\Gamma(\tfrac{1}{4})^2 \approx 0.478$ | 0.166 | 0.0886 | 0.0364 |
| 4 | $2\pi/48^{1/6}\Gamma(\tfrac{1}{3})^2 \approx 0.459$ | 0.112 | 0.0290 | $-0.00526$ |

that is adiabatically connected to the stationary state, *i.e.*, with $\lambda \to 0$ (for $s \to 0$), is the cumulant generating function [6]

$$\langle\!\langle N^j \rangle\!\rangle = \frac{\tau}{\tau_*}\frac{\partial^j}{\partial s^j}\lambda(\beta, sN_0)\Big|_{s=0} = \frac{\tau N_0^j}{\tau_*}\frac{\partial^j}{\partial t^j}\lambda(\beta, t)\Big|_{t=0}. \tag{6}$$

The form (5) of the effective Liouvillian directly implies what is known as data-collapse: if the time is measured in units of $\tau_*$ and the photon number in units of $N_0$, the results close to the bifurcation only depend on $\beta$, see Eq. (6) and also Fig. 3 as examples.

The problem below threshold for $\alpha = 0$ has been analyzed before [24, 29]. It has been found that

$$\langle\!\langle N^j \rangle\!\rangle = \frac{(2j-3)!!}{2}\frac{N_0^j}{|\beta|^{2j-1}}\frac{\tau}{\tau_*}. \tag{7}$$

In particular, the $j$-th cumulant has an apparent divergence like $\propto 1/|\beta|^{2j-1}$ when approaching the threshold from below. This result remains valid as long as $|\epsilon-\gamma|\tau_* \gg 1$. Sufficiently above threshold, on the other hand, the statistics is dominated by the Poissonian statistics of the coherent state with a (classical) photon current $\gamma|A_s/A_0|^2$.

## 4.2 Critical regime

Close to threshold with $\tau_*|\epsilon-\gamma| \lesssim 1$, we enter the critical regime where the nonlinearity becomes important, even though $\alpha \ll 1$. We find that, at threshold, the counting statistics is universal with the emerging timescale $\tau_* \propto \alpha^{-2/(m+2)}$ and the critical photon number $N_0 \propto \alpha^{-4/(m+2)}$ being the only parameters that depend on the microscopic details of the system. The cumulants are given by

$$\langle\!\langle N^j \rangle\!\rangle_{\text{thr}} = c_j \frac{N_0^j \tau}{\tau_*} \propto \frac{(2\bar{n}+1)^{[m-j(m-2)]/(m+2)}}{\alpha^{(4j-2)/(m+2)}}. \tag{8}$$

The universal numbers $c_j$ are listed in Table 1. Note that at threshold, the cumulants exhibit a universal scaling $1/\alpha^{(4j-2)/(m+2)}$ as a function of the nonlinearity $\alpha$. Moreover, we predict the universal ratio

$$r = \frac{\langle N \rangle_{\text{thr}}\langle\!\langle N^3 \rangle\!\rangle_{\text{thr}}}{\langle\!\langle N^2 \rangle\!\rangle_{\text{thr}}^2} = \frac{c_1 c_3}{c_2^2} = \begin{cases} 1.54, & m=2, \\ 1.05, & m=4. \end{cases} \tag{9}$$

of the first three cumulants at threshold, independent of the microscopic details of the system.

Of particular interest is the Fano factor $F = \langle\!\langle N^2 \rangle\!\rangle/\langle N \rangle = (c_2/c_1)N_0$ that allows to identify the parameter $N_0 \gg 1$ with the number of photons that are correlated at threshold. Note that for the generic case with $m=2$, we have $N_0 = f/\alpha$ such that $N_0$ is independent of the thermal occupation and only depends on the nonlinearity. The reduced Fano factor $F/N_0$ has

---

[6]More accurately, this is the factorial cumulant generating function. However, as we have $N_0 \gg 1$ the difference, being of order $1/N_0$, is small. Similarly, normal ordering is irrelevant in the limit of large photon numbers [29].

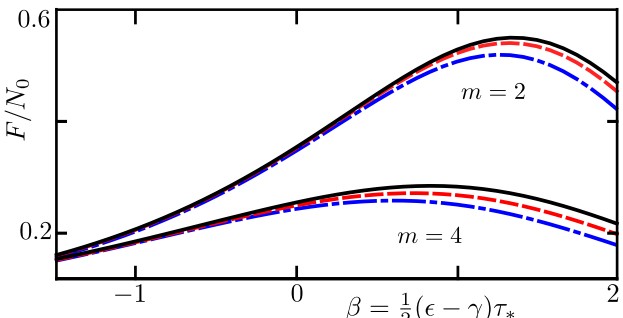

Figure 3: Comparison of the Fano factor of the effective model (5) (solid-line) with the full Lindblad evolution in the rotating frame (with $\bar{n} = 0$) (11) for the Josephson oscillator ($m = 2$) and the Duffing oscillator ($m = 4$) in the critical regime. The strengths $\kappa, \chi$ of the nonlinearities have been chosen such that $\alpha = 10^{-3}$ (dash-dotted line) and $\alpha = 10^{-4}$ (dashed line). It can be seen that the full models approach the results given by the universal Liouvillian in the limit $\alpha \to 0^+$. Note that almost all of the deviation can be attributed to a renormalization of the parameters in the effective model, lowering the threshold and decreasing $N_0$.

a universal form as a function of $\beta = \frac{1}{2}(\epsilon - \gamma)\tau_*$, see the solid lines in Fig. 3. We can see that the Fano factor becomes of order $N_0$ close to threshold. Interestingly, the maximal value does not occur at the threshold but rather at stronger driving. For $m = 2$, the maximal value is $F = 0.55f/\alpha$ (independent of temperature). It occurs at $\beta \approx 1.4$. For $m = 4$, the dependence of $F$ on $\beta$ is rather week and we obtain a relatively broad peak (with a maximum of $F = 0.28N_0$ at $\beta \approx 0.8$).

The timescale of the slow dynamics close to threshold is given by the correlation time $\tau_*$ that is connected to the width of the second-order coherence at threshold. In terms of the cumulants, it describes the ratio $F/I = (c_2/c_1^2)\tau^*$. The behavior of $g_{\text{thr}}^{(2)}$ for large $\tau$ is dominated by the eigenvalue $\lambda_2$ of $-\mathcal{L}_0$ (at $\beta = 0$) with the second smallest real part.[7] Using this fact, we find the following approximate form

$$g_{\text{thr}}^{(2)}(\tau) \approx 1 + \frac{\lambda_2}{2}\frac{F}{I}e^{-\lambda_2|\tau|} \tag{10}$$

of the second-order coherence, valid for $|\tau| \gtrsim \tau_*$; here, $\lambda_2 = 3.13/\tau_*$ for $m = 2$ and $\lambda_2 = 3.53/\tau_*$ for $m = 4$. When comparing to numerics, we find that the expression (10) is an excellent approximation with an error of less than a percent.

## 4.3 Comparison to numerics

In order to assess the validity of our results and connect them to possible experimental realizations, we simulate the time evolution of a parametrically driven, damped oscillator that exhibits a bifurcation. In the rotating-wave approximation, the system is described by the Lindblad master equation

$$\mathcal{L}_{\text{full}}(\rho) = -i[H, \rho] + \gamma(\bar{n} + 1)\mathcal{J}_a(\rho) + \gamma\bar{n}\mathcal{J}_{a^\dagger}(\rho) + sf\gamma a\rho a^\dagger, \tag{11}$$

with the jump operators $\mathcal{J}_L(\rho) = L\rho L^\dagger - \frac{1}{2}(L^\dagger L\rho + \rho L^\dagger L)$ and the Hamiltonian $H = \frac{i\epsilon}{4}(a^{\dagger 2} - a^2) + V(a^\dagger, a)$. We concentrate on two important nonlinearities $V$ that have been studied before. For a voltage-biased Josephson junction, previously studied, *e.g.*, in

---

[7]Due to the $\mathbb{Z}_2$-symmetry, there is no transition to the first excited state.

Refs. [22, 24–28, 40], the stabilizing potential assumes the form $V_J = \frac{i\kappa\epsilon}{48}(a^\dagger a^3 - a^{\dagger 3}a)$ with the effective fine-structure constant $\kappa = 16\pi Z_0 G_Q$, $Z_0$ the characteristic impedance of the oscillator, and $G_Q = e^2/\pi\hbar$ the quantum of conductance.[8] The system is driven by a voltage-biased Josephson junction with Josephson energy $E_J$. Using a method developed in [41], we can show that the slow dynamics of the system maps on the universal form (3) with $\epsilon = \kappa E_J/4\hbar$, $m = 2$, and $\alpha = \kappa/24$, see Appendix B for details. Typically, the characteristic impedance $Z_0$ is of the order of $50\,\Omega$ [40], which yields $\alpha \simeq 10^{-2}$, so indeed $\alpha$ is as small as required for the application of our theory. The Duffing oscillator with the nonlinearity $V_D = \chi\gamma a^\dagger a^\dagger aa$ that is independent of the phase of $a$ is the generic term for a system with a nonlinearity in the laboratory frame. In this case, due to the additional symmetry in the rotating frame, the nonlinearity leads to a weaker confining force [4]. As shown in Appendix B, the slow-dynamics close to thresholds maps on the universal form with $m = 4$ and $\alpha = 2\chi^2$. In Fig. 3, we compare the results of the full system to the ones of the effective model of the universal Liouvillian. As expected, we find rather good agreement in the whole critical regime with deviations controlled by the strength of the nonlinearity.

## 5 Conclusion

In conclusion, we have shown that in the critical regime, the dynamics of a degenerate parametric oscillator for long-time scales at weak nonlinarity $\alpha$ is described by a universal Liouvillian. The system only enters via the characteristic scales $\tau_*$ and $x_*$ that exhibit a universal dependence on $\alpha$. The effective model allows to efficiently calculate arbitrary observables. We have provided results for the photon current in the critical regime and for the first four cumulants at threshold. We predict that the ratio $\langle N \rangle_{\text{thr}} \langle\langle N^3 \rangle\rangle_{\text{thr}}/\langle\langle N^2 \rangle\rangle_{\text{thr}}^2$ of the first three cumulants assumes a universal value of order 1 at threshold. We have compared our results to the full numerical solution of the Lindblad evolution of a parametrically driven oscillator in the rotating frame. We have demonstrated that the effective model accurately describes the physics in the critical regime. In particular, we have shown that a voltage-biased Josephson junction embedded in a cavity, a system actively investigate in many experimental groups [40, 42–47], naturally realizes a weak nonlinearity. Thus, we believe that our results can be readily tested in present-day devices. Extending our approach to other driven-dissipative instabilities remains an interesting question for future research.

## Acknowledgments

F.H. thanks Yuli Nazarov for insightful discussions, especially during the initial phase of the project. This work was supported by the Deutsche Forschungsgemeinschaft (DFG) under Grant No. HA 7084/6-1.

## A Derivation of the universal Liouvillian

Using the general equivalence of the Langevin equation for a single stochastic path $x(t)$ and the Fokker-Planck equation for the corresponding probability-distribution $P(x, t)$ [37], Eq. (2)

---

[8]The Josephson energy in the rotating frame leads to the potential $iE_J : J_2(\sqrt{\kappa a^\dagger a})(a^{\dagger 2} - a^2)/2a^\dagger a :$ with $J_2$ a Bessel-function of the first kind. Expanding this result for $\kappa \ll 1$ yields the result quoted in the main text.

corresponds to the Fokker-Planck equation

$$\frac{\partial}{\partial t}P = \frac{\partial}{\partial x}\left[\frac{\gamma}{2}x - f(x) + \frac{\gamma(\bar{n}+\frac{1}{2})}{2}\frac{\partial}{\partial x}\right]P\,, \tag{A.1}$$

with $f(x) = \frac{1}{2}(\epsilon x - \gamma\alpha x^{m+1})$. Going over from $x$ to $q$ with $x = x_* q$, the Fokker-Planck equation can be brought into the dimensionless form

$$\frac{\partial}{\partial t}P = \frac{1}{\tau_*}\frac{\partial}{\partial q}\left[\frac{\tau_*(\gamma-\epsilon)}{2}q + q^{m+1} + \frac{1}{2}\frac{\partial}{\partial q}\right]P\,, \tag{A.2}$$

which corresponds to the universal Liouvillian (3) (via the identification $p = -i\partial/\partial q$).

## B  Derivation of the effective Lindbladian

Starting from the Lindbladian $\mathcal{L}_{\text{full}}$ given in Eq. (11) in the main text, we show how to derive the effective model describing the slow dynamics. The first step is to define a set of (super-)operators, $O_q = O_+ - O_-$ and $O_c = (O_+ + O_-)/2$ with $O_+(\rho) = O\rho$ and $O_-(\rho) = \rho O$ where $\rho$ is the density matrix of the system. Using these definitions for the ladder operators, we obtain the commutators $[a_q, a_c^\dagger] = [a_c, a_q^\dagger] = 1$ (while the rest of the commutators of $a_q, a_c, a_q^\dagger, a_c^\dagger$ vanish).

At first, we diagonalize the quadratic part of the Lindbladian $\mathcal{L}_{\text{quad}}$ for $s = 0$ and without the stabilizing potential in the Hamiltonian $H$ by a symplectic diagonalization to conserve the bosonic structure [41]. The Lindbladian assumes the form

$$\mathcal{L}_{\text{quad}} = -\tfrac{1}{2}(\gamma-\epsilon)v_s u_s - \tfrac{1}{2}(\gamma+\epsilon)v_f u_f \tag{B.1}$$

in terms of ladder operators $v_j, u_j$ such that $[u_j, v_k] = \delta_{jk}$. Explicitly, they are given by

$$u_s = x_c + i\frac{\gamma(2\bar{n}+1)}{2(\gamma-\epsilon)}y_q\,, \qquad v_s = -iy_q\,,$$
$$u_f = y_c - i\frac{\gamma(2\bar{n}+1)}{2(\gamma+\epsilon)}x_q\,, \qquad v_f = ix_q\,, \tag{B.2}$$

where we introduced the two quadratures $x_{q,c} = (a_{q,c}^\dagger + a_{q,c})/\sqrt{2}$ and $y_{q,c} = i(a_{q,c}^\dagger - a_{q,c})/\sqrt{2}$ with $[x_c, y_q] = [x_q, y_c] = i$. Note, however, that the creation and annihilation operators, $v_j$ and $u_j$, are not the adjoint of each other which is due to the fact that the Lindbladian $\mathcal{L}_{\text{quad}}$ is not Hermitian.

The slow dynamics of the system is given by the $u_s, v_s$ (or $x_c, y_q$ respectively) while $u_f, v_f$ correspond to a mode that decays with the rate $\gamma$ at threshold. The spectrum of the system is given by $\lambda = -\frac{1}{2}(\gamma-\epsilon)n_s - \frac{1}{2}(\gamma+\epsilon)n_f$ with $n_s, n_f \in \mathbb{N}_0$. The corresponding right eigenstates are the product states $|n_s, n_f\rangle = |n_s\rangle|n_f\rangle$ with $u_j|n_j\rangle = \sqrt{n_j}|n_j-1\rangle$ and $v_j|n_j\rangle = \sqrt{n_j+1}|n_j+1\rangle$. The left eigenstates similarly are characterized by $\langle n_j|v_j = \sqrt{n_j}\langle n_j-1|$ and $\langle n_j|u_j = \sqrt{n_j+1}\langle n_j+1|$. Note that in the superoperator formalism $|0,0\rangle$ corresponds to the stationary state $P_s$ and $\langle 0,0|$ is the trace operation.

Next, we can take the nonlinearity given by the stabilizing potential $V$ into account. It produces the additional term $\mathcal{L}_V = -i(V_+ - V_-)$ with $\mathcal{L}_{\text{full}} = \mathcal{L}_{\text{quad}} + \mathcal{L}_V + sf\gamma a_+ a_-^\dagger$. The small parameters $\alpha$ allows to treat $\mathcal{L}_V$ perturbatively. In particular, we are only interested in the effect of $\mathcal{L}_V$ on the slow mode as the fast mode remains in the stationary state $|n_f = 0\rangle$. Since we are close to threshold, we also set $\epsilon = \gamma$ in $\mathcal{L}_V$.

For the Josephson oscillator, we can simply project $\mathcal{L}_V$ onto the subspace with $n_f = 0$. Using the expressions (B.2), we obtain

$$\mathcal{L}_V \approx \langle n_f = 0|\mathcal{L}_V|n_f = 0\rangle = \tfrac{1}{16}\kappa\gamma i y_q x_c + \tfrac{1}{48}\kappa\gamma \left(i y_q x_c^3 + \tfrac{1}{4} i y_q^3 x_c\right). \tag{B.3}$$

The first term corresponds to a small shift of the threshold. This shift, which vanishes for $\alpha \to 0$, can be seen in Fig. 3. The model is then given by $y_q \mapsto p/x_*$ and $x_c \mapsto x_* q$ with a rescaling factor $x_* \gg 1$ that is specified in the main text. As discussed in the main text, we have $y_q \simeq x_*^{-1}$ and $x_c \simeq x_*$ and thus $y_q \ll x_c$ for $\alpha \to 0$. Due to this, the term $i y_q x_c^3$ is the most relevant nonlinearity. We can thus neglect $y_q^3 x_c$ which allows to identify $m = 2$ and $\alpha = \kappa/24$ for the Josephson oscillator.

The case of the Duffing potential needs more careful treatment due to the rotational symmetry as noted in the main text. In particular, we have $\langle n_f = 0|\mathcal{L}_V|n_f = 0\rangle = 0$ due to this symmetry. In order to find the stabilizing effect, we need to include virtual excitations of the fast mode in second-order perturbation theory. We obtain ($\bar{n} = 0$ for simplicity)

$$\mathcal{L}_V \approx \langle n_f = 0|\mathcal{L}_V|n_f = 0\rangle - \sum_{n_f' > 0} \frac{\langle n_f = 0|\mathcal{L}_V|n_f'\rangle \langle n_f'|\mathcal{L}_V|n_f = 0\rangle}{\gamma n_f'} \tag{B.4}$$

$$= \chi^2 \gamma \left(-\tfrac{31}{16} i y_q x_c - \tfrac{13}{4} i y_q x_c^3 + i y_q x_c^5 + \tfrac{5}{16} y_q^2 - \tfrac{5}{8} y_q^2 x_c^2 + \tfrac{1}{4} y_q^2 x_c^4 + \tfrac{13}{16} i y_q^3 x_c - \tfrac{5}{32} y_q^4 + \tfrac{1}{8} y_q^4 x_c^2 + \tfrac{1}{64} y_q^6\right).$$

As before, the rescaling $y_q \mapsto p/x_*$ and $x_c \mapsto x_* q$ leads to $y_q \ll x_c$ in the limit of weak nonlinearities. Because of this, the most relevant nonlinearity is the term $i\chi^2 \gamma y_q x_c^5$. This allows to identify $m = 4$ and $\alpha = 2\chi^2$ for the Duffing oscillator. Note that the term $\propto y_q x_c^3$ has the wrong sign and does not stabilize the mode. Moreover, it is subleading as $y_q x_c^3 \propto x_*^2$ while $y_q x_c^5 \propto x_*^4$. While the result (B.4) is shown for $\bar{n} = 0$ for simplicity, the relevant term $i\chi^2 \gamma y_q x_c^5$ is in fact independent of $\bar{n}$. The counting field leads to the additional term

$$sf\gamma a_+ a_-^\dagger = \frac{sf\gamma}{2}\left[(x_c + \tfrac{i}{2} y_q)^2 + (y_c - \tfrac{i}{2} x_q)^2\right]. \tag{B.5}$$

In the limit $\alpha \ll 1$, the dominant term is given by $\tfrac{1}{2} sf\gamma x_c^2 = \tfrac{1}{2} sf\gamma x_*^2 q^2$ as stated in the main text.

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
