# Peer review of "Radiation statistics of a degenerate parametric oscillator at threshold"

_SciPost Physics, doi:SciPost Phys. 14, 156 (2023)_

## Round 1 · Referee Report · Anonymous (Referee 1) · 2023-1-14

Strengths

  1. The paper addresses a timely topic, namely the scaling of fluctuations close to a threshold of a driven system. It sheds new light on the universal scaling of cumulants for such systems.

  2. The paper is well-written and mathematically rigorous and sound. It is also well-illustrated with figures.

  3. The comparison with the simulated time-evolution obtained from a Lindblad master equation at the end of the paper provides a nice comparison, and further substantiates the results of the paper.

Weaknesses

  1. The abstract should include at least a short description of the method used to obtain the results.

  2. The paper sometimes lacks a deeper physical interpretation of the mathematical results. For instance, how can we understand that the Fano factor reaches its maximum at a driving strength larger than threshold and not at the threshold itself? Is there a simple physical explanation/picture to understand this?

Report

The manuscript concerns the radiation statistics of a degenerate parametric oscillator at threshold, where both nonlinearity and fluctuations play a central role to the dynamics. The system is modeled using a Fokker-Planck equation, with a Liouvillian that also includes a source term accounting for the emitted photon current. From the eigenvalue adiabatically connected to the stationary state, the cumulant generating function of the emission statistics is obtained, fully describing the radiation statistics. The authors find that at threshold the counting statistics is universal, with the cumulants exhibiting a certain scaling as a function of the nonlinearity of the system. Furthermore, they identify a possibly universal ratio between the first three cumulants at threshold.

I find the paper interesting and of importance to further understand and investigate the properties of fluctuations at, or close to, thresholds/phase transitions in driven systems. In particular, the paper sheds new light on the universal scaling of cumulants of such systems, and thereby paves the way to further investigations on fluctuations at thresholds/phase transitions in various types of physical systems. As such, the paper fulfills the acceptance criterion on expectations (opening a new pathway in an existing research direction). Furthermore, I find the paper well-written and sufficiently detailed to make it possible to follow the authors’ arguments. The paper also contains sufficient citations and references to other relevant works. Finally, the conclusion is clear and well-motivated. Thus, in principle, the paper fulfills (almost) all acceptance criteria of SciPost Physics. However, before acceptance, the questions and comments stated in "Requested changes" should be properly addressed by the authors.

Requested changes

1) The abstract is not sufficiently detailed. At least, there should be a short mentioning of the method used to obtain the results. Otherwise, the paper fails to fulfill point 2 of the general acceptance criteria and cannot be published.

2) While the mathematics of the paper is presented in a sound and rigorous way, the authors could perhaps elaborate a bit more on the physical interpretation of the results. For instance, could the authors please comment on why the Fano factor reaches its maximum for a driving stronger than the threshold? Is this an expected/unexpected result?

3) Do the authors expect higher-order cumulants to fulfill universal ratios similar to the one that they identify for the first three cumulants [see Eq. (9)]? Why/why not?

4) Higher-order cumulants play an important role for determining Lee-Yang zeros and thus predicting the occurrence of phase transitions. Could the authors comment on the importance of their own results for further investigations on the behavior of Lee-Yang zeros for driven systems? What new insights about the physics at threshold can be gained from the results in this regard?

  • validity: top
  • significance: high
  • originality: high
  • clarity: high
  • formatting: excellent
  • grammar: perfect

Author:  Fabian Hassler  on 2023-02-14  [id 3356]

(in reply to Report 1 on 2023-01-14)
Category:
answer to question

We thank the referee for reviewing our work and for the positive feedback. In the following we will address the points of concern raised by the referee.

The referee writes:

1) The abstract is not sufficiently detailed. At least, there should be a short mentioning of the method used to obtain the results. Otherwise, the paper fails to fulfill point 2 of the general acceptance criteria and cannot be published.

Our response:

We have added two sentences to the abstract in order to explain our method.

2) While the mathematics of the paper is presented in a sound and rigorous way, the authors could perhaps elaborate a bit more on the physical interpretation of the results. For instance, could the authors please comment on why the Fano factor reaches its maximum for a driving stronger than the threshold? Is this an expected/unexpected result?

We would like to point out that close to the threshold both the nonlinearity as well as the quantum/classical fluctuations are important. In this fluctuation dominated regime, it is notoriously difficult to make a simple physical intuition work. The main point of our paper is a data-collapse; i.e., the numerical solution of the universal Liouvillian (Fig. 3) can be used to "predict" the Fano factor in all parametric systems close to the threshold. The microscopic details only enter via the scale factors $x_\ast, \tau_\ast$.

Nevertheless, we would like to offer to the referee the way we understand the fact that the maximum of Fig. 3 is for positive $\beta$: it is rather clear, that the Fano factor (starting at $F=2$ for low driving and approaching a coherent state with $F=1$ above threshold) has a maximum in between. A good description of the physics corresponding to the $\mathbb{Z}_2$ symmetry-breaking is a quantum particle in a double-well potential $-\beta x^2 + x^4$ where $\beta$ is tuned (see also the answer to point 4). Below the threshold ($\beta<0$), only a single minimum is present. Increasing $\beta$, the variance of the position increases (which corresponds to larger noise and Fano factor). However, the variance does not have a maximum at the threshold ($\beta=0$). Making $\beta$ slightly positive, the particle still gets more "room" to fluctuate as the confinement becomes weaker. Only at a larger value of $\beta$, the particle is confined to one of the two minima at finite $x_s$ and the variance (which is now around this minima) decreases.

3) Do the authors expect higher-order cumulants to fulfill universal ratios similar to the one that they identify for the first three cumulants [see Eq. (9)]? Why/why not?

There is an "infinite" number of other ratios that are also universal. We have given the one involving the lowest-cumulants as these are the easiest to measure. One can find other ratios from Eq. (8). The rule of the game is to come up with ratios where the factors $N_0$ and $\tau_*$ drop out. Other such ratios are for example $\frac{\langle N\rangle \langle\langle N^4\rangle\rangle}{ \langle\langle N^2 \rangle\rangle \langle\langle N^3 \rangle\rangle}= \frac{c_1 c_4}{c_2 c_3}$ or even $\frac{\langle N\rangle^2 \langle\langle N^4\rangle\rangle}{ \langle\langle N^2 \rangle\rangle^3 }= \frac{c_1^2 c_4}{c_2^3}$. Note that these other ratios involve higher-order cumulants.

4) Higher-order cumulants play an important role for determining Lee-Yang zeros and thus predicting the occurrence of phase transitions. Could the authors comment on the importance of their own results for further investigations on the behavior of Lee-Yang zeros for driven systems? What new insights about the physics at threshold can be gained from the results in this regard?

As far as we understand the theory of Lee-Yang zeros for open systems, the question is about the zeros of $Z(s) = \exp( \lambda(s) \tau/\tau_*)$. It is possible to map our Eq. (3) onto the Schrödinger equation of a (quantum) particle in a potential (see e.g. https://doi.org/10.1016/j.physa.2017.10.021). Using this mapping, we believe that we can show that $\lambda(s)$ remains finite for all $s\in\mathbb{C}$. The reason is that the potential, due to the finite nonlinearity, is confining for all $s$. Therefore, our result corresponds to the fact that $Z(s)$ is an analytical function without any zeros. The physical reason is that all the cumulants remain finite, due to the inclusion of the nonlinearity. The universal scaling of the cumulants (in the long-time limit at threshold) is then given by the scaling as a function of $\alpha$ as the cumulants diverge only for $\alpha \to 0$.

---

## Round 1 · Referee Report · Anonymous (Referee 2) · 2023-1-23

Report

The paper is devoted to a theoretical investigation of classical fluctuations in a parametric oscillator near the self-sustained oscillation threshold. In the linear approximation, the cumulants diverge at the threshold, and the authors include nonlinearity that bounds the cumulants to compute the corresponding values. To this end, they employ an effective FPE for slow nonlinear dynamics near the threshold. This equation is derived from the Lindblad equation for two realizations of the oscillator: voltage-biased Josephson junction and Duffing oscillator. The scaling parameters for the cumulants and characteristic time are presented as functions of the deviation from the threshold and nonlinearity coefficient.

The subject of the investigation is important, the methods employed are interesting, and the derivations I believe are technically correct.

However, an intricate way of presentation hides potential inconsistencies and raises questions.

The authors start by presenting a presumably general classical Langevin equation for parametric oscillators (1), which is specified in Eq (2), where the function f is defined. The authors comment that an integer parameter m in this function takes value 2 for a “generic oscillator” and value 4 for the Duffing oscillator. However, all the results in the paper are derived from Eq. (3), which is not a corresponding FPE but is an effective equation for slow dynamics derived from the quantum Lindblad equation (11). The derivation is not presented in the main text but is given in the Supplement. Here the authors explicitly write the Hamiltonian for two specific cases, voltage-biased Josephson junction and Duffing oscillator. One would assume, logically, that the former corresponds to a “generic oscillator” in (2) with m=2, and the latter to Duffing oscillator with m=4. However, this is not the case, both fall in the class m=2. A possible reason for this could be a different symmetry of the driving term in (11) compared to the common one due to the parametric driving of the oscillator frequency.

Furthermore, the presence of dissipative constant in the nonlinear terms in (11) and (2) indicates that the source of this nonlinearity is a multiphoton coupling to the environment rather than, commonly, a nonlinear energy of the oscillator. Whether this is taken care of in Eq. (11) remains unclear since the quantity L in the jump operator is never defined.

The authors should explain in the very beginning what physical oscillators are in mind.

An exact solution for a parametrically driven quantum nonlinear (Duffing) oscillator was derived in Opt. Commun. 127, 230, 1996 (not quoted). There the cumulants at the threshold were computed, and it would be useful to make a comparison with the results obtained in this paper.

It remained unclear to me whether Eq. (3) describes classical or quantum fluctuations. In the latter case, the results should refer to intrinsic fluctuations of the oscillator since the definition of the output photon flux adopted in the paper does not include the quantum correlations.

Summarizing, the paper meets the Expectation criterium 3: Opens a new pathway in an existing research direction, with clear potential for multipronged follow-up work. The paper meets all the Acceptance criteria except for (partially) item 3, see the report. The mentioned shortcomings must be explained in the revised manuscript
  • validity: -
  • significance: -
  • originality: -
  • clarity: -
  • formatting: -
  • grammar: -

Author:  Fabian Hassler  on 2023-02-14  [id 3355]

(in reply to Report 2 on 2023-01-23)
Category:
answer to question

We thank the referee for reviewing our work and for the overall positive feedback. In the following we will address the points of concern raised by the referee.

The referee writes:

The authors start by presenting a presumably general classical Langevin equation for parametric oscillators (1), which is specified in Eq (2), where the function f is defined. The authors comment that an integer parameter m in this function takes value 2 for a “generic oscillator” and value 4 for the Duffing oscillator. However, all the results in the paper are derived from Eq. (3), which is not a corresponding FPE but is an effective equation for slow dynamics derived from the quantum Lindblad equation (11).

Our response:

The referee has misunderstood this part of our manuscript by assuming that Eq. (3) is derived from Eq. (11). In fact Eq. (3) directly follows from Eq. (2) by the equivalence of a Fokker-Planck equation to its Langevin description. We have added a new appendix A to make this important point more clear and provide the necessary steps of the derivation. For the argument of the universality of the Liouvillian, it is crucial that (3) [as well as (2)] simply follow from symmetry arguments and the fact that the nonlinearity is weak.

The derivation is not presented in the main text but is given in the Supplement. Here the authors explicitly write the Hamiltonian for two specific cases, voltage-biased Josephson junction and Duffing oscillator. One would assume, logically, that the former corresponds to a “generic oscillator” in (2) with m=2, and the latter to Duffing oscillator with m=4. However, this is not the case, both fall in the class m=2. A possible reason for this could be a different symmetry of the driving term in (11) compared to the common one due to the parametric driving of the oscillator frequency.

We are afraid but we cannot follow the referee at this point. We never state that the Duffing oscillator corresponds to m=2. In fact it is stated at several points (the first being the first paragraph of the introduction) that the Duffing oscillator is not generic (with m=4) due to the additional symmetry. This result is not new but rather well-known in the literature, see e.g. Ref. [4].

Furthermore, the presence of dissipative constant in the nonlinear terms in (11) and (2) indicates that the source of this nonlinearity is a multiphoton coupling to the environment rather than, commonly, a nonlinear energy of the oscillator. Whether this is taken care of in Eq. (11) remains unclear since the quantity $L$ in the jump operator is never defined.

The fact that $\gamma$ appears in our expression of $f(x)$ is simply due to the fact that we would like to measure the nonlinearity with a dimensionless parameter $\alpha$. Close to threshold, we have $\epsilon \approx \gamma$ as the relevant bare frequency scale. We have added a new footnote 3 in the revised manuscript in order to explain this fact. Note that in Eq. (11) the (generic) jump operator $L$ is specified to $a$ and $a^\dagger$ as the terms describing the emission and absorption of photons.

The authors should explain in the very beginning what physical oscillators are in mind.

We have added two sentences at the end of the third paragraph of the introduction listing physical examples, where our results can be realized. The systems cover a broad range of physical implementations.

An exact solution for a parametrically driven quantum nonlinear (Duffing) oscillator was derived in Opt. Commun. 127, 230, 1996 (not quoted). There the cumulants at the threshold were computed, and it would be useful to make a comparison with the results obtained in this paper.

We would like to thank the referee for bringing the reference to our attention. However, we would also like to point out that the results of that paper, as far as we understand, only concern the stationary state inside the cavity. However, our results are valid for the photon-current leaving the cavity. For the latter, the dynamics of the system is important, not only the stationary state. In particular, the Fano factor of the photon-counting depends on the second-order coherence $g^{(2)}(\tau)$ for all times, whereas the paper mentioned by the referee only discusses $g^{(2)}(0)$. Because of this, it is not possible to compare the results to ours. Still, we have included the reference at the appropriate point in the introduction.

It remained unclear to me whether Eq. (3) describes classical or quantum fluctuations. In the latter case, the results should refer to intrinsic fluctuations of the oscillator since the definition of the output photon flux adopted in the paper does not include the quantum correlations.

Equation (3) describes both quantum and classical fluctuations. These correspond to $\bar n$ (classical) and $\frac12$ in Eq. (2). The fluctuations are included in a quasiclassical fashion that is possible for an oscillator with a weak nonlinearity (see Ref. [31] for details). The referee is correct that one has to be careful about proper normal ordering of the output photon flux in order to exclude vacuum photons in the counting. Note however that close to threshold the typical number of photons $N_0$ is large such that normal ordering is only an irrelevant correction. We have added a sentence to footnote 6 to explain this important point.

Summarizing, the paper meets the Expectation criterium 3: Opens a new pathway in an existing research direction, with clear potential for multipronged follow-up work. The paper meets all the Acceptance criteria except for (partially) item 3, see the report. The mentioned shortcomings must be explained in the revised manuscript

The revised manuscript has considerably improved the presentation of our results. We are confident that it now provides sufficient details such that the arguments and derivations can be reproduced.

---

## Round 2 · Referee Report · Anonymous (Referee 2) · 2023-2-23

Report

No more comments

---

## Round 2 · Referee Report · Anonymous (Referee 1) · 2023-2-27

Report

The authors have addressed my earlier comments properly. I recommend that the manuscript is accepted.

---

## Round 2 · Author Response

Dear Editor,
we thank the two referees for their favorable recommendation regarding the suitability of this work for SciPost Physics, and for their helpful suggestions for improvement of the manuscript.

---

## Round 2 · List of Changes

• We have included titles for the different sections

  • For better readability, we have placed the footnotes at the bottom of the pages rather than in the bibliography.

As reply to referee 1:

  • We have added two sentences in the abstract that explain our method.

As reply to referee 2:

  • We have added two sentences to the third paragraph in the introduction to highlight possible physical implementations, together with relevant references

  • We have added the new footnote 3.

  • We have added a sentence to footnote 6.

  • We have added the new appendix A that shows how to derive Eq. (3) from Eq. (2).

  • We have added the reference [30].

---

## Editorial Decision

published